# Polyphenols as a Diet Therapy Concept for Endometriosis—Current Opinion and Future Perspectives

**DOI:** 10.3390/nu13041347

**Published:** 2021-04-18

**Authors:** Agata Gołąbek, Katarzyna Kowalska, Anna Olejnik

**Affiliations:** Department of Biotechnology and Food Microbiology, Poznan University of Life Sciences, 48 Wojska Polskiego St., 60-627 Poznan, Poland; agata.golabek@up.poznan.pl (A.G.); katarzyna.kowalska@up.poznan.pl (K.K.)

**Keywords:** endometriosis, diet therapy, polyphenols, molecular targets, apoptosis, invasion, angiogenesis, inflammation, oxidative stress

## Abstract

Endometriosis represents an often painful, estrogen-dependent gynecological disorder, defined by the existence of endometrial glands and stroma exterior to the uterine cavity. The disease provides a wide range of symptoms and affects women’s quality of life and reproductive functions. Despite research efforts and extensive investigations, this disease’s pathogenesis and molecular basis remain unclear. Conventional endometriosis treatment implies surgical resection, hormonal therapies, and treatment with nonsteroidal anti-inflammatory drugs, but their efficacy is currently limited due to many side effects. Therefore, exploring complementary and alternative therapy strategies, minimizing the current treatments’ adverse effects, is needed. Plants are sources of bioactive compounds that demonstrate broad-spectrum health-promoting effects and interact with molecular targets associated with endometriosis, such as cell proliferation, apoptosis, invasiveness, inflammation, oxidative stress, and angiogenesis. Anti-endometriotic properties are exhibited mainly by polyphenols, which can exert a potent phytoestrogen effect, modulating estrogen activity. The available evidence derived from preclinical research and several clinical studies indicates that natural biologically active compounds represent promising candidates for developing novel strategies in endometriosis management. The purpose of this review is to provide a comprehensive overview of polyphenols and their properties valuable for natural treatment strategy by interacting with different cellular and molecular targets involved in endometriosis progression.

## 1. Introduction

Endometriosis is a chronic gynecological disorder, defined by the implantation of endometrial glands and stroma outside the uterine cavity, mainly in the pelvic peritoneum and ovaries. It affects approximately 10–15% of women of reproductive age, which extrapolates to around 190 million women worldwide [1,2]. Most affected patients often suffer from chronic pelvic pain, dyspareunia, dysmenorrhea, abnormal uterine bleeding, and infertility [3,4]. All these symptoms provide an impact on a patient’s quality of life, resulting in depression, anxiety, and impaired social function caused by the severity of pain [5]. The successful diagnosis of endometriosis requires surgical exploration or laparoscopy with histological confirmation [6]; therefore, the prevalence of the disease, its symptoms, associated disorders, and risk factors are limited data, only to the group of patients with definitive diagnosis [7].

Various hypotheses have been proposed to explain endometriosis pathology and tissue scattering throughout the abdominal cavity [8]. Distinct biological and clinical features indicate different types of endometriosis. Endometriosis can occur in several forms, according to histopathology and its anatomical localization in the pelvis: as the ovarian endometriotic cysts, deep infiltrating endometriosis, and superficial peritoneal lesions of varying colors [9]. Although scientists devoted significant effort to explain pathogenesis and endometriosis-associated pain, the basics of many biological processes remain enigmatic. Despite multiple theories explaining endometriosis pathogenesis, every concept suggests that endometriosis is a complex multifactorial and heterogeneous disease of uncertain etiology; its development and progression are influenced by genetic, immunological, hormonal, and environmental factors [10]. Recently, Lagana et al. (2019) summarized existing theories and divided them into two main categories: the transplantation and in situ [8]. Transplantation theory assumes that endometriosis develops due to eutopic endometrium metastasis to distinct sites by the hematogenous or lymphatic spread [8,11]. In contrast, in situ theory refers to endometriosis tissue existence caused by coelomic metaplasia, peritoneal mesothelium transformation into glandular endometrium or embryologic origin [8]. A separate theory does not explain the different clinical presentations and pathological forms in endometriosis. The formation and persistence of endometriotic tissues at ectopic sites essentially depend on the fundamental biological process as adhesion, proliferation, cell invasion, local inflammation, immune dysregulation, and neoangiogenesis [2]. Another increasingly reported important hallmark in endometriosis’s pathogenesis is the growth of new nerve fibers in endometriotic lesions, stimulated by inflammatory mediators and responsible for endometriosis-associated chronic pain conditions [12].

## 2. Current Treatment

Endometriosis therapy depends on the patient’s predominant symptoms, such as age, side-effect profile, the extent and location of endometriotic lesions, and previous treatment [13]. The main therapeutic approach includes surgery, pharmacotherapy, and long-term comprehensive individual treatment [14,15]. Surgical intervention represents a primary treatment aimed at destroying or removing ectopic endometriotic lesions. Indeed, surgery generally increases pain relief in some but not all women. Although the surgical resection option completely removes all visible ectopic endometriotic lesions, high recurrence rates of up to 50% within five years of surgery are reported [16].

The most common pharmacological approach for endometriosis therapy is nonsteroidal anti-inflammatory drugs for managing pain symptoms, combined with oral contraceptives [17,18]. Hormone therapy aims to induce hypoestrogenemia, inhibition of tissue proliferation, and inflammation [19]. Second-line treatments that suppress systemic estrogen levels include progestin monotherapy and gonadotrophin-releasing hormone agonists (GnRH) [20]. The efficacy of conventional medical treatments is limited or intermittent in most patients. It brings a plethora of side effects like perimenopausal stage symptoms, osteoporosis, breakthrough bleeding, lipid profile changes, and liver dysfunction, resulting from the hypo-estrogenic state induced by this medical approach [21,22]. Consequently, exploring additional and alternative options is needed to improve treatment outcomes for patients with endometriosis. Strategies complement conventional drug therapy may include non-pharmaceutical options: acupuncture, diet changes, supplementations, and phytotherapy [23,24,25]. Natural therapies constitute an option of paramount importance that can accelerate the healing process and help minimize the current treatment’s adverse effects. The search for new alternative therapies should start with demonstrating their benefits and safety, evidence-based core outcomes obtained using endometriotic models [26].

## 3. Nutritional and Dietary Aspects of Endometriosis

Because of the unsuccessful treatment and chronic character of endometriosis, many women affected by endometriosis use additional management strategies to control this disease themselves [27]. According to the last Australian national online survey, as many as 76% of endometriosis women use non-pharmacological practices and lifestyle choices like relaxation techniques, movement, and nutrition. Almost half the women managed dietary supporting, and the diet’s effectiveness had high self-reported improvement scores [28]. In recent years, an increasing number of endometriosis patients have focused on health-promoting and therapy-supporting dietary factors [29].

Plants are promising sources of bioactive compounds, potentially improving therapeutic strategies [30]. Different pathways involved in the physiological and pathological processes associated with endometriosis, including altered inflammatory microenvironment, the attachment and invasion mechanisms, angiogenesis, estrogen activity, menstrual cyclicity, organochlorine burden, and prostaglandin metabolism, can be a target of food bioactive compounds [15,29,31]. Dietary treatment of endometriosis may be based on the estrogen dependency of endometriosis, and estrogen-lowering diet components may be used to treat or regress endometriosis [29,32]. Changing dietary patterns for endometriosis may help reduce inflammatory markers, shown to be increased in endometriosis [33]. The diet to treat endometriosis can moderate prostaglandins’ effects responsible for the pain during endometriosis progression [34]. Different active compounds offering various therapeutic properties such as antiproliferative, anti-inflammatory, antioxidant, and analgesic properties are considered a complex group of molecular compounds effective in endometriosis [35].

There are many scientific studies on the effects of nutrition on endometriosis. Most papers have reported case-control studies evaluating the role of diet on endometriosis risk, while the diet effectiveness and diet compound potential in endometriosis treatment have investigated less frequently. In our opinion, a scientific basis for the action of bioactive compounds in the management of endometriosis should be considered, and their possible curative effect described to explain their contribution to the healing process and impact on the control of severe endometriotic symptoms.

## 4. Molecular Targets in Endometriosis Dietary Management

Plants are sources of natural bioactive compounds that demonstrate broad-spectrum health-promoting effects and interact with molecular targets associated with endometriosis, such as cell proliferation and apoptosis, cell adhesion, invasiveness, inflammation, oxidative stress, and angiogenesis. The anti-endometriotic potential of nutraceuticals can also concern a potent phytoestrogenic effect modulating estrogen activity. Dysregulated physiological processes related to endometriosis, contributing potential molecular targets for bioactive polyphenol compounds, are schematically presented in Figure 1.

### 4.1. Cell Survival and Apoptosis

The proliferation of endometrial cells is primarily controlled by interactions between sex steroids and their corresponding receptors. Endometriotic tissue’s proliferative potential is significantly higher in the eutopic endometrium of women with endometriosis than in the endometrium of disease-free patients [36]. The complex microenvironment consists of proinflammatory and endocrine mediators in surrounding endometriotic lesions, promoting endometriotic cells’ proliferation [37]. Ectopic endometriotic tissues are characterized by reported markedly higher levels of estrogen receptor beta (ERβ) compared with eutopic endometrial tissues and cells. ERβ plays a critical role in anti-apoptosis signaling and is responsible for a mechanism of evasion from endogenous immune surveillance for cell survival through inactivation of TNFα-induced apoptosis complex I and II and the apoptosome. ERβ also activates the cytoplasmic inflammasome components, resulting in increasing interleukin 1β (IL-1β), enhancing adhesion and proliferation of endometrial cells. [38]. Furthermore, proliferation is enhanced by high levels of estrogen in the microenvironment of endometriotic lesions provided by locally increased expression of aromatase and decreased 17β-hydroxysteroid dehydrogenase (17β-HSD) type 2 in endometriotic implants [9].

Nuclear factor kappa B (NF-κB), a pleiotropic transcription factor, also plays a critical role in developing endometriosis by protecting cells from apoptosis via activating anti-apoptotic genes and inducing the proliferation of the endometriotic cells [39]. Under normal conditions, endometrial cells from healthy women during menstruation do not survive in ectopic location because cell turnover is regulated by apoptosis, avoiding cell dissemination and attachment. Endometrial tissue’s impaired susceptibility to apoptosis leads to increased invasiveness and the abnormal survival of endometrial cells in the peritoneum [40]. Different studies suggest that dysregulated expression of specific proteins associated with apoptosis, including B-cell lymphoma 2 (BCL-2) protein family, B-cell lymphoma-extra large (BCL-XL) protein, BCL-2 associated X (BAX) protein, FAS/FASL system, cysteine-aspartic proteases (caspases), and survivin represent possible factors involving in apoptosis resistance in endometriotic cells [41,42,43]. Some studies have evaluated the effect of different bioactive food compounds on mechanisms regulating endometriotic cells proliferation and apoptosis, suggesting their therapeutic potential.

### 4.2. Cell Attachment and Invasion

The initial attachment of refluxed endometrial tissue fragments to the pelvic peritoneum is the principal of Sampson’s retrograde theory of endometriosis origin [44]. Sub-peritoneal endometriotic lesions’ establishment requires remodeling of the extracellular matrix of peritoneal mesothelium and invasion in their surrounding environment [45]. Menstrual effluent and morphological alterations can easily damage the intact mesothelium—a protective barrier against the implantation; the own adhesion site can also be created to implant regurgitated endometrial cells [46,47,48]. The adhesion of endometrium fragments to the peritoneum in women with endometriosis is enhanced by the overproduction of cellular adhesion molecules that facilitate intercellular binding and cellular attachments with the extracellular matrix; including CD44 transmembrane glycoprotein, cell adhesion molecules (CAM) such as integrins, cadherins, selectins, the immunoglobulin superfamily (Ig-CAM), and transmembrane-anchored proteoglycans like syndecans [49,50].

The progression of the invasion of adjacent tissues requires extracellular matrix degradation. The breakdown and remodeling of extracellular matrix thought to be mainly modulated by matrix metalloproteinases (MMP), especially MMP-1, 2, 3, 9, and 11 and their inhibitors (tissue inhibitors of metalloproteinases, TIMP) [51]. MMPs are the initial mediators of maintenance and survival of endometriotic lesions, mainly that their expression enhances significantly in endometriotic implants [52]. Different hormones, inflammatory cytokines, including IL-6, IL-1, and growth factors, regulate MMPs. The primary inhibitor is progesterone, which might indirectly regulate MMP expression through the plasminogen activator pathway, increasing the levels of plasminogen activator inhibitor 1 (PAI-1) and reducing the activation of latent MMP-proenzyme by plasmin [53].

### 4.3. Angiogenesis

The formation of new blood vessels is fundamental to creating and maintaining endometriotic lesions, especially in the peritoneal microenvironment, indicating the importance of new blood supply in endometriosis development. The local neovascularization is augmented by the complex mixture of growth factors, proangiogenic factors, steroid hormones, inflammatory cytokines present in peritoneal fluid [54]. Many pro-angiogenic factors, including fibroblast growth factors (FGF), platelet-derived endothelial cell growth factor (PDGF), transforming growth factor alpha and beta (TGF-α and TGF-β), hepatocyte growth factor (HGF), erythropoietin, angiogenin, tumor necrosis factor alpha (TNF-α), and IL-8 have been detected at increased concentrations in peritoneal fluid of women with endometriosis [54]. Vascular endothelial growth factor (VEGF), produced in high amounts by activated peritoneal macrophages, neutrophils, lymphocytes, and endometrial stromal cells, is considered the most crucial angiogenic agent to induce proliferation and migration of endothelial cells, that can also increase vascular permeability [55,56]. VEGF is exerted in response to physiological activators, such as inflammation and hypoxia. Bone-marrow-derived endothelial progenitor cells detected in early-stage of endometriotic lesions are recruited by hypoxia-inducible-factor1-alpha (HIF-1α) and stromal-cell-derived-factor 1 (SDF-1) [57,58]. New agents, like antiangiogenic factors, could constitute a new and promising approach to treating this disease [59].

### 4.4. Immune Dysregulation

Inflammation has a vital role in the progression of endometriosis and is associated with altering immune cell profile in the peritoneal cavity [33]. The endometrial tissue is a significant source of inflammatory mediators like cytokines, chemokines, and prostaglandins, which attract macrophages, neutrophils, monocytes, eosinophils, and T cells, that can affect processes in the abdominal cavity [60]. The peritoneal fluid of endometriosis is characterized by the enhanced concentration of inflammatory mediators like IL-1, IL-6, IL-8, TGF-β, TNF-α, VEGF, cyclooxygenase-2 (COX-2), and monocyte chemoattractant protein 1 (MCP-1) [61,62]. Changes in inflammatory mediators result from the aberrant function of almost all types of immune cells in women with endometriosis.

Peritoneal macrophages are the most prevalent immune cells, found in the highest quantity in the peritoneal fluids. The peritoneal environment is regulated by the activated macrophages, which can scavenge cellular debris, and remove red blood cells and damaged tissue fragments. Lesion resident macrophages can induce tissue repair, inflammation, and angiogenesis through the secretion of soluble mediators like cytokines, prostaglandins, and enzymes. However, in the peritoneum of endometriosis-affected women, macrophages have a decreased phagocytic activity, and the amount of regurgitated endometrial cells in the peritoneal cavity may be higher. The imbalance in M1 and M2 macrophages was reported in endometriosis with an upregulation of the M2 type. M2 macrophages are supposed to have a crucial role in endometriosis development by modulating adaptive immune response and, consequently, promoting the implantation and proliferation of endometrial cells [33]. Macrophage-derived factors such as prostaglandins and cytokines produced in the peritoneal cavity, may also modulate natural killer cells activity [63]. The cytotoxic function of NK cells against endometrial cells reaching the peritoneal cavity is decreased and is inversely related to the advanced stages of the disease [64]. Moreover, reduced T cells’ cytotoxic activity, modulation of pro-inflammatory cytokine secretion and altering Th1 cytokine profile prevalent in the early stages of the disease, to a Th2 cytokine profile in late stages and autoantibody production by B lymphocytes were observed [8,65]. The presence of this altered inflammatory niche could favor the implantation and development of endometriotic lesions from refluxed endometrial tissue. There is substantial evidence that the immune system plays a crucial role in this disease. Novel treatment strategies targeting immune pathways are urgently needed [13].

### 4.5. Oxidative Stress

Oxidative stress occurs due to an imbalance between reactive oxygen/nitrogen species (ROS/RNS) production and the organism’s ability to scavenge and detoxify ROS/RNS harmful effects [66]. Essential molecules in the body, such as membrane lipids, nucleic acids and proteins, are ROS targets. Oxidative stress is involved in the pathogenesis of many chronic diseases, such as cancer, neurodegenerative and cardiovascular diseases, and ageing [67]. Endometriosis may involve ROS production because macrophages, erythrocytes, and apoptotic endometrial tissue, transplanted into the peritoneal cavity through retrograde menstruation, are found to be inducers of oxidative stress. The accumulation of iron in the different components of the peritoneal cavity of endometriosis patients has reported. Peritoneal iron overload may be a consequence of hemoglobin breakdown or bleeding of peritoneal lesions. The macrophages and lymphocytes are recruited and activated by releasing the proinflammatory heme products and the oxidative stress signals. Enhanced activity of peritoneal macrophages and lymphocytes, in turn, strengthen the oxidation stress in the endometriotic peritoneum [68]. Moreover, ROS activates NF-κB, which leads to the expression of multiple genes engaged in cell growth, angiogenesis, and inflammation mechanisms regulation in the endometriosis [69]. A subclass of nitrogen-containing compounds can play detrimental effects in endometriosis pathology. In women with endometriosis, the increased concentration of endothelial and inducible forms of nitric oxide synthase (eNOS and iNOS), the enzymes that ultimately produce nitric oxide (NO), has been found in the peritoneal fluid. In endometriosis condition, the elevated levels of NOS and NO are correlated with impaired reproductive biological processes, e.g., ovulation, fertilization, implantation, and embryonic development. Moreover, the increased endometriosis-related eNOS activity has shown a positive correlation between estrogen and progesterone levels [70]. Additionally, aberrant change in the potential of endogenous antioxidant enzymes like superoxide dismutase (SOD) and glutathione peroxidase (GPX) can contribute to the oxidative damage that occurs in endometriosis [70]. Estrogen participates in the SOD induction to overcome the excessive oxidative stress; however, overexpression of this antioxidant enzyme has adverse effects and contributes to promoting endometriotic cell survival and protecting them from apoptosis. Superoxide anion produced in large amounts mediates several signaling pathways and contributes to endometriosis-related pain through activation of nociceptive neurons [71]. Oxidative stress has been suggested as an etiological factor of chronic pelvic pain, and antioxidant supplementation has a proven impact on reducing pain in endometriosis patients [72]. These findings highlight the importance of developing therapeutic approaches targeting oxidative imbalance: reducing the oxidative stress status may exert protective effects against endometriosis.

### 4.6. Hormonal Imbalance

Hormonal imbalance in endometriosis is the master regulator of the alternations of multiple cellular functions, such as proliferation, invasion, angiogenesis, inflammation, and neurogenesis and pain generation [55]. Endometrial tissues are controlled by steroid hormones like 17β-estradiol and progesterone, which change the expression of hundreds of genes during different phases of the menstrual cycle. Eutopic endometrial tissues and endometriotic tissues in ectopic locations contain the immunoreactive estrogen receptors and progesterone receptors (PR); therefore, they respond to 17β-estradiol and progesterone with apparently similar histological changes [73]. Local estrogen production in both the ectopic and eutopic endometrium is considered to stimulate tissue’s growth and play a crucial role in regulating the immunological mechanisms responsible for controlling the development of endometriosis. The enzyme aromatase, a member of the cytochrome P450 superfamily, is responsible for the last step in the synthesis of estradiol through the aromatization of androgens (androstenedione and testosterone) into estrogens (estrone and estradiol, respectively) [74]. In premenopausal women with endometriosis, estradiol arises from three major tissue sites that express aromatase. These are ovaries, which primarily convert cholesterol to estradiol and secrete it periodically into the circulation, peripheral tissues such as the adipose tissue and skin, that convert androstenedione to estrone in relatively small quantities, and endometriotic tissue, synthesizing *de novo* high amounts of estradiol from cholesterol via aromatase and steroidogenic acute regulatory protein (StAR) [73]. Contrary to endometriotic lesions, normal endometrium cannot synthesize estrogen due to these enzymes’ absence [75]. In endometriosis, local estradiol levels increase due to upregulation of estradiol-producing aromatase expression and reduction in 17β-HSD type 2 activity, implicated in the inactivation of estradiol through the oxidation to less active estrone [76]. In normal conditions, the expression of 17β-HSD type 2 in epithelial cells is activated by paracrine signaling triggering by the progesterone via PRs on stromal cells. There are two isoforms of PRs: PR-A and PR-B; PR-B likely plays a more crucial biological function in the endometrium. Progesterone resistance in endometriotic tissue is due to the total reduction in PRs and a drastic downregulation of PR-B in endometriotic stromal cells [77]. In the case of estrogen receptors (ER), a significant reduction in isotype ERα and an increase in ERβ is observed. The elevated expression of ERβ is associated with its promoter hypomethylation, while the decrease in ERα is due to its promoter hypermethylation and direct inhibition by ERβ [78]. Increasing the E2/ERβ ratio is considered to enhance lesion survival and inflammation. It is linked with the positive feedback loop by stimulation of COX-2 activation, which is involved in producing hormones, such as prostaglandin E_2_ (PGE_2_), with significance in inflammation and pain. In turn, PGE_2_ influences steroidogenic genes, mainly overexpressing aromatase, and supports the sustained production of estradiol [78,79]. Novel alternative treatment to regulate estradiol biosynthesis and modulate its coactivators without suppressing ovulation, instead of conventional pharmacotherapy, could be designed to more beneficially control this disorder. The substances that bind competitively to ERs, such as phytoestrogens, are recommended for future research assessment to demonstrate their effect on the levels of hormones and inflammatory markers in endometriosis.

## 5. Potential of Polyphenol Compounds in Endometriosis Management

There are increasing investigations about the beneficial effect of bioactive compounds in preventing chronic non-communicable diseases, including cardiovascular diseases and cancers. Several studies have evaluated the relationship between polyphenols’ administration, such as flavonoids and phytoestrogens, and female cancer risk. Grosso et al. (2016) observed that regular consumption of diets rich in flavonoids is associated with decreased breast and ovarian cancer risk [80]. In the meta-analysis, Micek et al. (2020) have found that higher dietary intake of phytoestrogens (i.e., isoflavones) is inversely correlated with the risk of breast cancer mortality and recurrence [81]. Endometriosis and breast cancer are estrogen-dependent conditions and share similar cellular processes like proliferation, invasion, neo-angiogenesis, metastases, and reproduction-associated risk factors. Significant similarities between endometriosis and cancer encourage investigating the protective effect of natural bioactive molecules against this disease [82].

Moreover, endometriosis has been linked to chronic systemic inflammation, oxidative stress, and an atherogenic lipid profile, leading to initiating and maintaining cardiovascular diseases [83]. The meta-analysis showed the linear relationship between increased dietary intake of total flavonoids and lower risk of cardiovascular disease [84]. Similar dietary patterns rich in bioactive food compounds may exert beneficial effects toward endometriosis.

Specific plant-derived bioactive compounds, exhibiting anti-proliferative, anti-oxidant, anti-inflammatory, anti-angiogenic, pro-apoptotic, immunomodulatory, and estrogen modulating activities, can affect different pathways involved in endometriosis’s etiopathogenesis and provide new strategies for treating or regressing endometriosis [85,86]. Numerous preclinical research and clinical trials in endometriosis studies have attributed to phytochemicals a broad range of beneficial biological activities and described molecular mechanisms and pathways through which they may effectively influence disease progression [68,87]. The studies on potential therapeutic phytochemicals had mostly focused on polyphenolic compounds.

Polyphenols comprise a large group of bioactive compounds synthesized by plants that are an integral part of the human diet, widely recognized for their multiple functional health-promoting benefits, mainly anti-oxidant and anti-inflammatory properties, with importance in endometriosis [88]. Phenol-Explorer, the first comprehensive database on polyphenol content in foods (http://phenol-explorer.eu/) (accessed on 12 April 2021), classifies polyphenols into six primary groups: flavonoids, lignans, non-phenolic metabolites, other polyphenols, phenolic acids, and stilbenes. Flavonoids constituting the largest group of dietary polyphenols have been divided into nine subgroups, including anthocyanins, chalcones, dihydrochalcones, dihydroflavonols, flavanols, flavanones, flavones, flavonols, and isoflavonoids (http://phenol-explorer.eu/) (accessed on 12 April 2021). The above classification has been applied to organize structurally different polyphenol compounds as promising candidates for developing novel strategies in treating endometriosis. This article describes polyphenols’ molecular action profiles, which target fundamental disease processes’ complex pathophysiology. Table 1 summarizes polyphenols’ mechanisms of action and molecular targets in endometriosis management.

### 5.1. Flavonoids

#### 5.1.1. Flavonols

##### Quercetin

Quercetin is a naturally occurring flavonol found in various fruits and vegetables such as onions, cauliflower, apples, berries, and chili peppers [89]. Quercetin exerts the anticancer effects by reducing cancer cell viability and enhancing apoptosis and autophagy through the modulation of Akt, mammalian target of rapamycin (mTOR), and HIF-1α signaling pathways [130]. An in vitro study demonstrated that quercetin inhibits cell proliferation and induces cell cycle arrest and apoptosis in endocervical cell lines VK2/E6E7 and End1/E6E7. Cell apoptosis symptoms included DNA fragmentation, loss of mitochondrial membrane potential and downregulation of extracellular signal-regulated kinase 1/2 (ERK1/2), p38 mitogen-activated protein kinase (p38 MAPK), and AKT signaling molecules [89].

Antiproliferative and anti-inflammatory effects of intraperitoneally injected quercetin were observed in an auto-implanted mouse model of endometriosis. A decrease in cyclin D1 expression in the luminal and glandular epithelium was observed in response to quercetin injection [89]. In rats with induced endometriosis, quercetin administration significantly decreased endometrial implants’ size and serum E2 and TNF-α level compared to the related control group. The results indicated that quercetin treatment significantly increased antioxidant NAD(P)H enzyme quinone oxidoreductase 1 (NQO1) activity and its major transcription factor nuclear factor erythroid 2-related factor 2 (Nrf2) expression. The study depicted that quercetin, unlike conventional treatments of endometriosis, does not undesirably affect estrogen and progesterone production, emphasizing an advantage of this compound over the currently used medications. Furthermore, regression of endometrial lesions was found as an effect of enhanced autophagy through mTOR inhibition and overexpression autophagy-related markers, including Beclin-1 and Atg5, after the combined treatment with quercetin and metformin [90].

Favorable effects of natural active ingredients affecting main pathophysiological pathways of endometriosis demonstrated in preclinical models require proof in clinical trials. Signorile et al. (2018) investigated 30 patients defined as stage IV of endometriosis treated for three months with a dietary supplement containing, e.g., 200 mg quercetin. Clinicians assessed the reduction in the inflammatory response and endometriosis’s symptoms and its harmful effects on affected organs in patients with endometriosis. Experimental data obtained proved a significant effect in reducing the pain and other disabling disease symptoms in the patients treated with a dietary supplement. The substantial decrease in serum PGE_2_ levels observed after treatment indicated supplement anti-inflammatory potential. Moreover, in treated patients, the endometriosis foci’ size was reduced due to the decreased serum CA-125 concentration, as the authors believe. The authors suggest that further studies of dietary supplement combined with standard therapies are necessary to establish potentially practical new endometriosis treatment strategies [91].

#### 5.1.2. Flavones

##### Apigenin

Scientists suggest that apigenin (4′,5,6-trihydroxyflavone), present in significant amounts in various fruits (apples, grapes, oranges) and vegetables (onion, parsley, celery), is suitable for the prevention and treatment of diverse diseases such as diabetes, metabolic disorders, cardiovascular and neuronal diseases, and cancers [131]. Apigenin prevents or inhibits disease progression by the involvement in cell cycle arrest, apoptosis induction, anti-inflammatory, and free radical scavenging mechanisms; thus, it is also proposed as a potential therapeutic agent in reproductive disorders [92]. In this context, apigenin reduced the proliferation of two human endometriotic cell lines in a dose-dependent manner and induced apoptosis by inhibition of phosphorylation of ERK1/2 and c-Jun N-terminal kinase (JNK) proteins and increasing pro-apoptotic proteins, including BAX and cytochrome c. Additionally, apigenin treated cells accumulated excessive ROS, experienced lipid peroxidation and endoplasmic reticulum stress and lost mitochondrial membrane potential with an increase in calcium ions in the cytosol [92]. In another study, apigenin reduced mitogenic activity, inflammatory response, and attenuated TNF-α–induced cell proliferation and COX-2/PGE_2_ synthesis via the inactivation of the NF-κB pathway in endometriotic stromal cells [93]. The biological potential of apigenin derivatives-rich extracts obtained from *A. austriaca* flowers has been shown in a rat endometriosis model. The most active apigenin-rich fraction remarkably decreased adhesion and endometriotic implant volumes and TNF-α, VEGF, and IL-6 levels [94]. Consistent with apigenin’s biological activity results, this flavone is suggested as a potential novel therapeutic agent to overcome current limitations in preventing or treating endometriosis. However, further preclinical basic research and clinical trials, in particular, are strongly required.

##### Baicalein

Baicalein (5,6,7-trihydroxyflavone) is a flavonoid derived from *Scutellaria baicalensis* Georgi’s root, a traditional herb in Chinese herbal medicine [132]. Baicalein’s pharmacological properties, such as anti-tumor, anti-bacterial, anti-viral, anti-allergic, anti-oxidant, and cytoprotective effects, have been reviewed recently [95]. Anti-cancer effects, target mechanisms and signaling pathways of baicalein are of increasing interest from scientists. Baicalein has been studied in many types of malignancy, including bladder, melanoma, prostate, and breast cancer, as well as in gynecologic tumors such as cervical and ovarian cancer [133]. Baicalein strongly inhibited the proliferation of breast cancer MCF-7 cells and induced apoptosis via suppressing 17β-estradiol-induced transactivation [134]. Other in vivo studies revealed the inhibitory effect of baicalein against cervical cancer [135]. Molecular mechanisms described so far include regulation of cell cycle by inhibition several cyclins or cyclin-dependent kinases (CDK), cell proliferation arresting through the Wnt/β-catenin signaling pathway, suppression of PD-L1 expression to promote anti-tumor immunity, as well as induction of apoptosis and autophagy via the ERK/phosphatidylinositol-4,5-bisphosphate 3-kinase (PI3K)/Akt pathway and caspase activation [136,137,138]. Due to the reported anti-tumor properties, Jin et al. (2017) and Ke et al. (2021) evaluated baicalein anti-endometriotic potential using human endometrial stromal cells in vitro. The results indicated that baicalein reduces ectopic endometrial stromal cell viability and induces apoptosis through the NF-κB signaling pathway. Following baicalein treatment, the level of anti-apoptotic protein BCL-2 decreased significantly. The number of cells in the G1 phase increased with a simultaneous decline in the number of cells in the S and G2/M phases compared with non-treated control cells [95]. More recent findings indicate that baicalein reduces ectopic endometrial stromal cell invasiveness, as demonstrated by analysis of invasion-related proteins, including MMP-9, MMP-2 and membrane type 1-matrix metalloproteinase (MT1-MMP) [96]. In a mouse model of endometriosis, intraperitoneal administration of baicalein inhibited ectopic lesions’ growth, reduced MT1-MMP, proprotein convertase of MMPs (FURIN) and TGFB1 expression. Thus, baicalein with a potent pro-apoptotic and anti-invasive activity may provide a novel treatment strategy for endometriosis, as the authors suggested [96]. Promising preclinical research results indicate that undertaking clinical studies on endometriosis treatment with baicalein is warranted and recommended to obtain strong evidence for therapeutic baicalein potential.

##### Wogonin

Wogonin (*O*-methylated flavone), another flavonoid compound found in *Scutellaria baicalensis* Georgi root extract, has been reported in preclinical studies as an anti-tumor agent [97]. An anti-proliferative effect of wogonin has been analyzed in human endometrial stromal T-HESC cell line and primary stromal cell cultures established from endometrial biopsies from women with endometriosis. In these in vitro models, wogonin significantly inhibited primary and T-HESC cell proliferation causing cell cycle arrest at the G2/M phase and attenuated the ERα expression. In a murine model of endometriosis, wogonin reduced lesion size and decreased the number of cells in the proliferation state, simultaneously increasing the number of apoptotic cells [97].

#### 5.1.3. Isoflavonoids

##### Daidzein and Genistein

Isoflavones, a subgroup of the flavonoids found in soybeans, share a similar structure with estradiol but reportedly display antagonistic estrogen properties [22]. Structural similarities due to their heterocyclic phenolic structure facilitate isoflavones bind to estrogen receptors ERα and ERβ and mimic estrogenic actions [139]. Soy isoflavones are found mainly as glycosides with daidzein and genistein are the major representative isoflavone aglycones abundant in soybeans, which are converted by β-glycosidase derived from bacteria that colonize the small and large intestine [140,141].

A few in vitro and in vivo studies have investigated the effect of isoflavones on endometriosis. Daidzein-rich isoflavone aglycones (DRIA) inhibited cell proliferation of the endometriotic stromal cells at clinically relevant concentrations. DRIAs led to reduced expression of IL-6, IL-8, COX-2, and limited aromatase activity, as well as decreased levels of glucocorticoid-regulated kinase and PGE_2_ in serum. DRIAs also reduced the number, weight and Ki-67 proliferative activity of endometriosis-like lesions in the murine endometriotic model [142]. Another study showed that genistein causes regression of an endometriotic implant in a rat model. In the presence of estrogen, genistein has demonstrated antagonistic estrogen activity on endometriotic implants [98]. The results from animal studies indicate DRIAs as a potential therapeutic agent for improving treatment strategies in endometriosis.

Clinical research showed the reduced risk of endometriosis development following soy isoflavones’ ingestion. A significant correlation between higher urinary isoflavone levels (genistein and daidzein) and decreased risk of advanced endometriosis (stage III and IV), but not early endometriosis (stage I and II) occurrence, was observed within a case-control study involving Japanese women [99]. Future clinical studies and meta-analyses focusing on the benefits of phytoestrogen-rich product consumption by endometriosis patients are strongly required.

##### Puerarin

Puerarin is the primary bioactive ingredient isolated from the root of the *Pueraria lobata*. It has proven potential in managing cardiovascular diseases, cerebrovascular disorders, cancer, Parkinson’s disease, Alzheimer’s disease, diabetes, and diabetic complications [143]. Puerarin exerted a vasodilatory effect by activating large conductance voltage and calcium-activated potassium channels in rat models. Puerarin could significantly inhibit diabetic vascular complications by decreasing P-selectin, low-density lipoproteins, and cholesterol in the blood and downregulating mRNA expression of aorta vascular cell adhesion molecule in diabetic rats. The preventive activity of puerarin against oxidative stress-induced neurodegeneration has been attributed to blocking ROS overproduction and lipid peroxidation, inducing nuclear Nrf2 protein expression, and activating catalase and GSH peroxidase and the transcription and translation levels of heme oxygenase 1 [143].

Puerarin is commonly regarded as a phytoestrogen because it binds to estrogen receptors and competes with estradiol to exhibit an anti-estrogenic effect [144]. Its role in treating endometriosis, an estrogen-dependent disorder, has been well elucidated in several studies. Puerarin suppressed invasion and vascularization of endometriosis tissue activated by E2 through reducing the accumulation of MMP-9, ICAM-1, and VEGF and increasing the level of tissue inhibitor of metalloproteinase 1 (TIMP-1) in primary endometriotic stromal cells and in chick chorioallantoic membrane in vivo model [100]. Puerarin can suppress estrogen-stimulated proliferation of endometriotic stromal cells and downregulate the transcription and protein level of cyclin D1 and cdc25A by promoting ERα corepressors’ recruitment and limiting the recruitment of coactivators. The investigation suggests that a puerarin-dependent interaction between ERα and corepressor complexes may be pivotal for puerarin’s anti-estrogenic effect, indicating that this isoflavone could be a potential therapeutic compound for endometriosis treatment [101]. In vivo investigations showed that puerarin inhibits aromatase and COX-2 in ectopic endometrium, thereby reducing E2 and PGE_2_ and consequently prevents the growth of ectopic endometriotic cells [102]. Clinical studies on the potential of puerarin in the treatment of endometriosis have not been conducted to date.

#### 5.1.4. Flavanones

##### Naringenin

Naringenin, mainly found in citrus fruits, is known for having anti-oxidant, anti-proliferative, anti-inflammatory, and anti-angiogenic properties in chronic and metabolic diseases [145,146]. In cancers, as preclinical studies reported, naringenin induced ROS-mediated cell death and inhibited cell migration and invasion [146,147]. Naringenin being a phytoestrogen, impairs ERα signaling by interfering with MAPK, PI3K, and ERK1/2 signaling pathways [148]. Considering the biological potential, scientists undertook investigations to evaluate the ameliorative role of naringenin in endometriosis.

In vitro studies showed that naringenin suppresses proliferation and induces apoptosis in endometriosis cells by decreasing mitochondrial membrane potential and ROS generation. The apoptotic effect of naringenin involved activation of PI3K and MAPK cell signaling pathways [103]. An animal study revealed naringenin potential against endometriosis by reducing expression of various endometriosis prognostic markers (TAK1, PAK1, VEGF, PCNA, MMP2, and MMP-9) in endometriotic lesions developed in rats. Naringenin prevented endometriosis propagation via anti-inflammatory mechanisms by modulating the serum TNF-α, NO level, and proapoptotic effects as ROS overproduction and the loss of mitochondrial membrane potential. Further, naringenin provoked depletion of Nrf2, a transcriptional factor controlling transcription of endogenous antioxidant enzymes, which protect cells against oxidative damage and consequent down-regulation of cytoprotective genes, thereby inhibiting endometriotic cell proliferation [104]. The main direction for future studies indicated by scientists concerns the continuation of research on naringenin potential in endometriosis treatment by applying endometrium samples from a larger population of endometriotic patients [104].

#### 5.1.5. Chalcones

##### Xanthohumol

Xanthohumol, a prenylated flavonoid, originates from naringenin chalcone and found in hop’s female inflorescences (*Humulus lupulus* L.) [149]. Numerous studies have demonstrated its pleiotropic chemopreventive activity for cancer due to anti-proliferative, anti-inflammatory, and anti-angiogenic properties [149]. Xanthohumol effectively inhibited endometriotic lesions’ development and reduced the level of PI3K protein in a BALB/c mouse model of endometriosis. Moreover, xanthohumol decreased the microvessels’ density and did not adversely affect the reproductive tract organs, suggesting its potential for the selective treatment of endometriosis [105].

#### 5.1.6. Flavanols

##### Epigallocatechin Gallate

Epigallocatechin gallate (EGCG) is the main flavanol found in black and white tea, especially in green tea [150]. This compound has received enormous pharmacological attention because of its putative beneficial health effects for treating different cancer types, based on its anti-oxidant, anti-angiogenic, and anti-proliferative properties [151,152].

Ex vivo studies showed that EGCG suppressed cell proliferation and induced cell death in endometrial epithelial cells from human biopsies [119]. Interestingly, EGCG more effectively inhibited cell proliferation of endometriotic stromal cells than their normal counterparts, such as endometrial stromal cells [152]. Studies performed by Matsuzaki and Darcha (2014) revealed that treatment with EGCG significantly inhibited proliferation, migration, and invasion of endometrial and endometriotic stromal cells via inhibition of MAPK and Smad signaling pathways. EGCG also attenuated cell-mediated collagen gel contraction according to both endometriotic and endometrial cells. EGCG treatment substantially decreased the expression of markers αSMA, collagen-I, fibronectin (FN), and connective tissue growth factor (CTGF) known to be involved in fibrogenesis in endometriotic stromal cells [106]. Another study demonstrated that treatment with EGCG for two weeks reduced endometriotic lesions by downregulating angiogenic VEGFA mRNA expression and upregulating NF-κB and MAPK 1 mRNA expression [153]. The subsequent experiments showed that EGCG exerted an inhibitory effect on endometriosis-associated angiogenesis by decreasing the number and size of microvessels and VEGF-C/VEGF receptor 2 (VEGFR-2) expression and signaling [107]. A pro-drug of EGCG (pro-EGCG, EGCG octaacetate) with improved stability and bioavailability exhibited higher antioxidant, antiangiogenic, and inhibitory activity than EGCG in the mice model. Treatment with the pro-EGCG resulted in poor lesion neovascularization and decreased plasma VEGF concentration [108]. EGCG, which exerts potent anticancer effects, could represent an anti-angiogenic agent for endometriosis.

Currently, the Chinese University of Hong Kong has started a clinical trial to evaluate green tea’s efficacy and safety in endometriosis. In the study, 185 patients with endometrioma will randomize into either experimental group, which will receive high-purity EGCG (400 mg of SUNPHENON EGCG) twice per day or a placebo group. Supplement administration will take three months before a planned surgery. Change in endometriotic lesion size will be a primary measured outcome of the study. Further, monitoring pain, quality of life, endometriotic lesions collected as biopsies, the number and density of neovasculature and severe adverse events and side effects are scheduled as the secondary outcomes (Clinical Trials.gov ID: NCT02832271).

### 5.2. Other Polyphenols

#### Curcumin

Curcumin is a polyphenolic monomer extracted from turmeric of *Curcuma longa* L. Scientific research suggests that curcumin represents some potential therapeutic roles as an anti-inflammatory, anti-cancer, and anti-aging agent [154]. Several in vitro and in vivo studies have proved its pharmacological activities in endometriosis management.

In human endometriotic stromal cells, curcumin attenuated TNF-α-stimulated expression of ICAM-1, VCAM-1, and secretion of IL-6, IL-8, and MCP-1 by inhibiting activation of the transcription factor NF-κB [109]. Additionally, curcumin reduced endometrial cell proliferation by decreasing estrogen level [110]. Moreover, treatment with curcumin altered human endometriotic and endometrial stromal cell morphology, and interfered with cell proliferation and cell division. Curcumin arrested the cell cycle in the G1 phase and induced apoptosis via downregulation of VEGF expression in human endometriotic and endometrial stromal cells [111]. A recent study has shown that curcumin is a potent inhibitor of secretion of pro-inflammatory and pro-angiogenic chemokines and cytokines by stromal cells derived from eutopic endometrium in endometriosis and by normal endometrial stromal cells [112].

In the murine model, curcumin administration was associated with the reduction in endometriosis progression and apoptosis activation. Curcumin caused the regression of endometriosis predominantly via the cytochrome c-mediated mitochondrial pathway of apoptosis, and apoptotic responses include both p53-dependent and independent path. Curcumin acted as an inhibitor of NF-κB translocation and downregulated MMP-3, thus mediating endometriosis regression [113]. Besides, curcumin therapy in endometriosis mice demonstrated a protective role against lipid peroxidation and protein oxidation [114]. In experimental rat models, curcumin decreased the weight and volume of endometriotic tissues in a dose-dependent manner via downregulating VEGF expression [115]. Curcumin inhibited endometriotic foci development by interfering with microvessels’ density, showing potent antiangiogenic activity [116]. In summary, curcumin targets multiple molecular and cellular mechanisms in the pathogenesis of endometriosis, such as inflammation, attachment and angiogenesis, that might considerably improve and relieve endometriosis management.

Interestingly, the Medical University of Vienna recruited endometriosis patients for a randomized interventional clinical trial of a curcumin dietary supplement called Flexofytol, administered in two capsules containing 42 mg of curcumin twice a day for four months. Clinicians assumed that the average pain score from baseline to four months after treatment begins would be a primary outcome measured. As the secondary outcomes, they defined a decrease in the number of days with pain, alleviation of dyspareunia, dysuria, dyschezia, and the change in the quality of life and sexual function (Clinical Trials.gov ID: NCT04150406).

### 5.3. Phenolic Acids

#### Rosmarinic Acid

Rosmarinic acid, found in several plants such as *Rosmarinus officinalis*, *Salvia officinalis*, and *Thymus* sp., is a phenolic compound with various health-promoting and therapeutic properties [155,156]. Its antioxidant, anti-inflammatory, anti-tumor, and anti-angiogenic functions may be relevant for endometriosis treatment. Ferella’s team (2018) assessed the anti-endometriosis potential of rosmarinic acid and carnosic acid (phenolic diterpene), which occur together in rosemary (*Rosmarinus officinalis*) and common sage (*Salvia officinalis*). They applied human endometrial stromal cell culture and a BALB/c model with induced endometriotic-like lesions. In the study, both compounds suppressed cell proliferation and reduced the size of endometriotic lesions in mice. Furthermore, rosmarinic acid promoted apoptosis in endometriotic tissue and reduced intracellular ROS accumulation in human endometrial stromal cells [97].

### 5.4. Stilbenes

#### Resveratrol

Resveratrol is a phytoalexin polyphenol, which various plants naturally produce in response to ultraviolet radiation and fungal infections [157]. Resveratrol naturally occurs in many plant species, including peanuts, berries, legumes, and grasses; however, resveratrol’s richest sources are grapes and red wine. The first scientific reports about the resveratrol phenomenon presented in 1992 indicated resveratrol as a cardioprotective agent addressed to the French paradox and initiated extensive research activity on this compound [158]. Besides its cardioprotective effects, resveratrol has become well known due to its anti-cancer potential, first demonstrated a few years later, as appeared by its ability to suppress all carcinogenesis stages [159]. Ever since, numerous studies suggest that resveratrol has various health beneficial properties, including anti-cancer, anti-inflammatory, anti-oxidant, anti-microbial, and anti-angiogenic functions [160]. It can also protect against many age-related diseases, such as cardiovascular disease, diabetes, arthritis, and neurodegenerative disorders [161].

Several possible molecular targets for the protective effects of resveratrol have been proposed. COX-2, a key regulator of tumorigenesis development and circulatory homeostasis, has been inhibited by resveratrol in the in vitro and in vivo models of cancers. Cardiovascular benefits of resveratrol refer to its property to facilitate endothelial NO production, preventing thrombogenic and atherogenic processes by relaxing vascular smooth muscle cells and upregulating blood flow [161]. Resveratrol’s capacity to extend lifespan underlies activation of silent mating type information regulation homolog 1 (SIRT1), a class III histone deacetylase responsible for controlling gene expression, DNA repair, metabolism, response to oxidative stress, mitochondrial function, and biogenesis. SIRT1 protects cells from oncogenic transformation and acts on specific targets in defined cellular signaling pathways in tumors. Resveratrol has been reported to possess neuroprotective, anti-tumorigenic functions via SIRT1 activation [162].

Resveratrol is widely considered an innovative natural agent in the alternative and complementary therapy of severe conditions by affecting multiple pivotal transduction pathways involved in disease progression. Resveratrol anti-inflammatory potential, including prostaglandin synthesis inhibition, has been suggested to probably contribute to the prevention and treatment of endometriosis [163]. The therapeutic effect of resveratrol on endometriosis was first evaluated by Bruner-Tran et al. (2011). They reported that resveratrol could reduce the number and size of endometriotic lesions in a nude mouse model. The resveratrol effect correlated with decreased endometrial cell proliferation and increased apoptotic cell death in lesions. In an in vitro study, resveratrol induced a concentration-dependent reduction in endometrial stromal cell invasiveness [117]. Since that report, interest in resveratrol has grown significantly, including resveratrol’s protective mechanisms in endometriosis.

In the next research, resveratrol reduced human endometrial epithelial cell proliferation and increased apoptosis in primary cultures [119]. Compared to estrogen, resveratrol acted as both an agonist at low concentrations, whereas it played an antagonistic role at high concentrations. In the immunocompromised mouse model of endometriosis, reduced cell proliferation in the lesion has been observed with concomitantly decreasing epithelial ERα level [118]. Moreover, a recent study revealed that resveratrol treatment reduced expression of IGF-1 and HGF, pivotal molecules in endometriotic lesion growth and angiogenesis, in eutopic and ectopic endometrial stromal cells derived from endometriosis patients [120]. Another recent study looked at the effect of resveratrol at different concentrations on tissue growth, angiogenesis and NO secretion, and expression of apoptosis-related genes in human endometriotic and endometrial tissue in 3D culture. The mean growth of endometriotic and endometrial tissue showed significant dose-dependent inhibition during the treatment with resveratrol. Furthermore, resveratrol dose- and time-dependently reduced NO concentration in endometriotic and endometrial tissues. Experimental data also demonstrated that the expression of proapoptotic genes P53, BAX, CASP3, and SIRT1 increased significantly in the endometriotic and endometrial tissues after treatment with various resveratrol doses [121]. In vitro study showed the anti-inflammatory effect of resveratrol reflected in the suppression of MCP-1, IL-6, IL-8, and RANTES in ectopic endometrial stromal cells of endometriotic women [122].

Several promising consistent in vivo studies reported the beneficial effects of resveratrol on endometriotic lesions in animal models. Namely, resveratrol attenuated the inflammatory response in the peritoneal environment and decreased endometriosis development through its anti-angiogenic and anti-inflammatory properties. Resveratrol inhibitory effect was evaluated using an experimentally induced endometriosis rat model. Results showed a significant reduction in the implant size, a decrease in VEGF and MCP-1 in the peritoneal fluid and VEGF in plasma, and highly suppressed of VEGF expression in the endometriotic tissue within considerable histological changes in the endometriotic foci following 14-day treatment with resveratrol at a dose of 10 mg/kg [123]. In a similar rat model with induced endometriosis, resveratrol’s effectiveness was evaluated based on its ability to inhibit angiogenesis and inflammation. After a 21-day administration, resveratrol in a 60 mg/kg dose reduced the implants’ areas and lowered VEGF and MCP-1 levels. Interestingly, resveratrol’s therapeutic potential was comparable with leuprolide acetate, used as a drug in conventional endometriosis treatment [124]. In another study in the rat endometriosis model, resveratrol showed ameliorative effects on endometriotic implants due to its potent antioxidant properties. After treatment, scientists observed a dose-dependent reduction in endometriotic implant volumes and increased SOD and GPX activities in rats’ sera and tissues [125]. In a more recent study, Kong et al. (2020) presented the role of epithelial‒mesenchymal transition in endometriosis’s pathogenesis and resveratrol suppressive potential within metastasis-associated pathways in endometriosis. Resveratrol inhibited the proliferation, migration, and invasion of endometrial stromal cells in in vitro cultures, and suppressed ectopic endometrium growth in the mouse endometriosis model. Kong’s team observed that resveratrol suppressed the epithelial-mesenchymal transition process in endometriosis mice and endometrial cells. In these experimental models, resveratrol downregulated the expression of metastasis-associated protein 1 (MTA1) and transcription factor zinc finger E-box-binding homeobox 2 (ZEB2), essential components that contribute to metastasis by promoting the transformation of epithelial cells into mesenchymal cells [126].

Clinicians also investigated the therapeutic potential of resveratrol in endometriosis in several clinical trials. Most clinical studies generally presupposed resveratrol’s capacity to potentiate the actions of oral contraceptives in the treatment of endometriosis through its hypo-estrogenic action [164]. Maia et al. (2012) assessed the advantages of the association of resveratrol with oral contraceptives for the management of endometriosis-related pain. They monitored the resveratrol effect in 12 patients who failed to obtain pain relief after treatment with an oral contraceptive containing drospirenone with ethinylestradiol. After two months of using 30 mg of resveratrol to the standard hormone therapy, 82% of patients reported complete resolution of dysmenorrhea and a significant reduction in pain occurrence. In a separate experiment, the same authors investigated resveratrol’s potential for regulating COX-2 and aromatase expression, as reducing these enzymes is significant in overcoming chronic pelvic pain. Clinicians analyzed the aromatase and COX-2 expression in the endometrial tissue of 42 patients, of which 26 patients administered the combination of resveratrol with an oral contraceptive. The results showed that the combined therapy successfully reduces the aromatase and COX-2 expression compared with hormone therapy alone [127].

Another clinical trial investigating the use of resveratrol as an adjuvant treatment of pain assumed a randomized, double-blind, controlled comparison of resveratrol versus placebo to treat endometriosis-related pain. The study involved 44 eligible women with a laparoscopic diagnosis of endometriosis. It investigated whether 40 mg/day of resveratrol with a monophasic contraceptive pill could attenuate pain scores after 42 days of administration. The result presented no difference between treatments when comparing pain scores between groups after treatment with resveratrol plus contraceptive and contraceptive with placebo. The secondary outcomes of this research measured the CA-125 and prolactin level. The obtained data documented the reduction in CA-125 plasma level in both the placebo and resveratrol groups but did not find changes in prolactin level [165].

More recently, researchers at the Tehran University of Medical Sciences carried out an exploratory clinical trial, which included 34 patients with endometriosis-associated infertility. Participants of the study, randomly and equally divided into control and treatment groups, took 400 mg resveratrol twice daily for 12–14 weeks along with oral contraceptives in the last three weeks. The clinical trial showed that resveratrol could modify the inflammation process in the endometrium of women with endometriosis. The results obtained revealed downregulation of the mRNA and protein expression of MMP-2 and MMP-9 in endometrium tissue, endometrium fluid, and serum following resveratrol treatment. In conclusion, the authors emphasized the need for further research on the effect of resveratrol on MMP-2 and MMP-9 expression in women with endometriosis [128].

The same group of scientists examined the effect of resveratrol on VEGF and TNF-α expression in the eutopic endometrium of infertile patients with endometriosis within the implantation window in a randomized exploratory trial. As previously, the study involved 34 patients, divided into a placebo-controlled group and treatment group of 17 patients administrated 400 mg of resveratrol for 12 to 14 weeks. The obtained results indicated that resveratrol decreased VEGF and TNF-α expression as markers of angiogenesis and inflammation. According to the authors, further clinical trials are necessary to reveal resveratrol involvement in different pathways like angiogenesis, inflammation, and oxidative stress to confirm resveratrol potential as a therapeutic approach in endometriosis treatment [129].

## 6. Conclusions and Future Perspectives

The pathophysiologic mechanisms involved in developing an extremely heterogeneous disease, like endometriosis, have not yet been fully elucidated, and available treatment interventions are currently limited. The discovery of new therapeutic agents and improvements in existing treatment strategies seems to be necessary. Natural polyphenols exhibit a pleiotropic action profile and could exert anti-endometriotic effects via comprehensive interactions with multiple molecular targets associated with endometriosis, such as cell proliferation, apoptosis, inflammation, oxidative stress, angiogenesis, and invasiveness. Moreover, some of these phytochemicals can exert a potent phytoestrogenic effect modulating the estrogen networks without inducing severe side effects contrary to the conventional anti-estrogenic therapy against endometriosis. The available results of some studies reveal that natural bioactive compounds do not affect fertility, reproductive organs, and the development of offspring [105,108]. Additionally, an alternative endometriosis therapy approach with therapeutic potential similar to conventional management is inexpensive and suitable for long-term and safe endometriosis patients’ treatment.

However, existing evidence suggesting an attractiveness of naturally occurring substances for treating endometriosis has been provided mainly by pre-clinical in vitro research and animal studies. Despite several clinical trials performed, there is still insufficient proofs to recommend implementing natural treatment strategies in daily clinical applications. Besides, detailed pharmacokinetic and pharmacodynamic data on the administration of many polyphenols are still unavailable; thus, further clinical studies are necessary to confirm the efficacy of these compounds in the treatment of endometriosis. Another significant limitation is the lack of physiological relevance of high polyphenol doses not achievable at routine food consumption, usually applied in the experimental endometriosis in vitro models. Moreover, most in vitro studies showed the therapeutic potential of naturally occurring polyphenols without considering their gastrointestinal digestion and metabolism by gut microorganisms. The extensive gastrointestinal metabolism of polyphenol compounds influences their bioavailability and intestinal absorption and, consequently, their therapeutic efficacy in tissues and organs. Furthermore, in combining natural agents with conventional pharmacotherapy, studies on bioactive compound-drug interactions are mandatory. Knowledge of the precise mechanisms of action of the natural compounds in the endometriosis’s network of several cell signaling pathways is required, and further studies on this topic are needed.

It is worth emphasizing that polyphenols’ overconsumption, especially as highly concentrated supplements isolated from the food matrix, may cause harmful health effects [166]. Reduced thiamin and folic acid transport have been reported due to polyphenols’ pre-absorptive interactions during digestion in the gastrointestinal tract. Similarly, significant changes in some drugs’ bioavailability can be observed due to polyphenols’ interaction with drug transporters or digestive enzymes [167]. Besides, polyphenols can exert an iron-chelating effect and reduce iron absorption, which leads to deterioration of iron status. Previously published research hypothesized that soy-derived isoflavones adversely affect patients with endometriosis and estrogen-sensitive breast and endometrial cancer due to estrogen-like biologic activities and hormone-interfering properties [168,169]. However, most recent clinical trials have not confirmed the adverse effects of isoflavones and other phytoestrogens on women with risk and active estrogen-dependent cancers [170,171]. According to the latest reports, isoflavone-rich food and supplements are safe dietary factors for peri- and post-menopausal women and endometriosis patients [99,172].

In our opinion, natural-based endometriosis therapies are unlikely to serve as an exclusive therapy strategy and exert a potent curative effect, but they can overcome endometriosis-related symptoms. Combined with precision medical methods individually tailored to adjust patients’ needs, the natural compounds can constitute an integral part and a central direction of developing future endometriosis therapeutic concepts.

## Figures and Tables

**Figure 1 nutrients-13-01347-f001:**
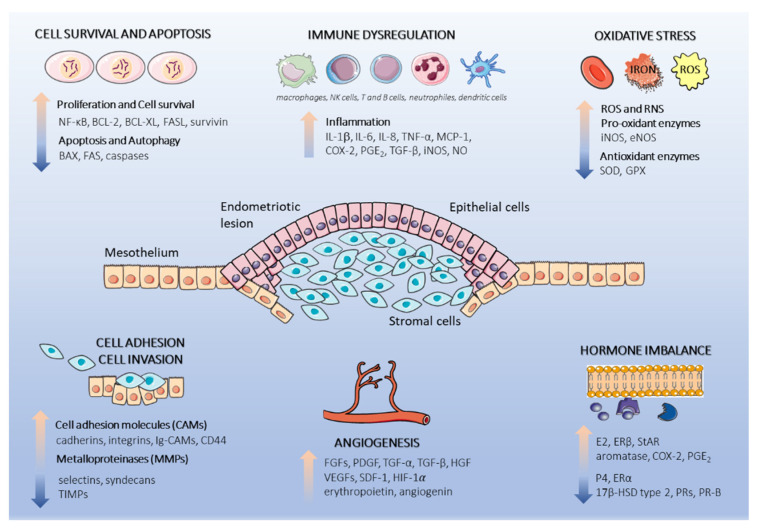
Schematic representation of dysregulated physiological processes in the endometriotic lesion. Endometriotic lesions consist of epithelial and stromal endometrial cells originating form retrograde menstruation. The endometrial cells subsequently attach to the underlying peritoneal mesothelium, proliferate, and initiate pro-endometriotic microenvironment formation. Cell proliferation and resistance to apoptosis, cell adhesion, inflammation, oxidative stress, and hormone signaling are augmented by the interactions of multiple cells and secretion of cytokines, chemokines, and hormones, which facilitate endometriotic lesion progression. This figure was made using graphic components obtained from the website: www.servier.com/powerpoint-imagebank (15 March 2021). Abbreviations used in graphics: 17β-HSD type 2, 17β-hydroxysteroid dehydrogenase type 2; BAX, BCL-2 associated X protein; BCL-XL, B-cell lymphoma-extra large; BCL-2, B-cell lymphoma 2; COX-2, cyclooxygenase-2; eNOS, endothelial nitric oxide synthase; ERα, estrogen receptor alpha; ERβ, estrogen receptor beta; E2, estradiol; FAS, tumor necrosis factor receptor superfamily member 6; FASL tumor necrosis factor ligand superfamily member; FGFs, fibroblast growth factors; GPX, glutathione peroxidase; HGF, hepatocyte growth factor; HIF-1α, hypoxia-inducible-factor 1-alpha; IL-1β, interleukin 1β, IL-6, interleukin 6; IL-8, interleukin 8; iNOS, inducible nitric oxide synthase; MCP-1, monocyte chemoattractant protein 1; NF-κB, nuclear factor kappa B; NO, nitric oxide; PGE_2_, prostaglandin E_2_; PR-B, progesterone receptor isoform B; P4, progesterone; ROS, reactive oxygen species; RNS, reactive nitrogen species; SDF-1, stromal-cell-derived-factor 1; StAR, steroidogenic acute regulatory protein; SOD, superoxide dismutase; TGF-α, transforming growth factor alpha; TGF-β, transforming growth factor beta; TIMP-1, tissue inhibitor of metalloproteinase-1; VEGF, vascular endothelial growth factor.

**Table 1 nutrients-13-01347-t001:** Natural polyphenol compounds and their mechanisms of action and molecular targets analyzed in endometriosis preclinical and clinical studies.

Mechanisms of Action	Molecular Targets	Disease Model	Ref.
**Quercetin**			
↓Cell proliferation ↑Apoptosis	↓Mitochondrial membrane potential, ↓ERK1/2, ↓p38 MAPK, ↓AKT, DNA fragmentation	Endometriosis cell lines	[89]
↓Cell proliferation	↓*CCND1*, ↓Cyclin D1	Murine endometriosis model	[89]
↓Cell proliferation, ↑Autophagy ↓Endometriotic lesion size ↓Oxidative stress	↓Serum E2, ↓Serum TNF-α ↑NQO1 enzyme activity, mTOR inhibition ↑ *NRF2*, ↑*BECN1*, ↑*ATG5*	Rat endometriosis model	[90]
↓Inflammation ↓Endometriosis-related pain ↓Endometriotic lesion size	↓Serum PGE_2_ ↓Serum CA-125	30 patients with IV endometriosis stage treated for 3 months with 200 mg quercetin	[91]
**Apigenin**			
↓Cell proliferation ↑Apoptosis	ERK1/2 and JNK inhibition, ↑BAX, ↑Cyt- c, ↑ROS, ↑ER stress, ↑Calcium ions in cytosol ↓Mitochondrial membrane potential	Endometriosis cell lines	[92]
↓Cell proliferation ↓Inflammation	↓COX-2, ↓PGE_2_, ↓IL-8 NF-κB inhibition	Primary endometriotic stromal cells from ovarian endometrioma	[93]
↓Angiogenesis, ↓Inflammation ↓Endometriotic implant volume	↓Peritoneal VEGF, ↓Peritoneal TNF-α ↓Peritoneal IL-6	Rat endometriosis model	[94]
**Baicalein**			
Cell cycle arrest, ↓Cell viability ↑Apoptosis	NF-κB inhibition, ↑Cells in the G1 phase ↓Cells in the S and G2/M phases, ↓BCL-2	Endometrial stromal cells from patients with ovarian endometriosis	[95]
↓Cell invasiveness	↓MMP-9, ↓MMP-2, ↓MT1-MMP, ↓TGFB1, ↓FURIN	Ectopic endometrial stromal cells	[96]
↓Cell invasiveness ↓Endometriotic lesion growth	↓MT1-MMP, ↓FURIN, ↓TGFB1	Murine endometriosis model	[96]
**Wogonin**			
↓Cell proliferation Cell cycle arrest	↑Cells in the G2/M phase, ↓ERα	Endometrial stromal T-HESC cells Primary endometriotic stromal cells	[97]
↓Endometriotic implant size ↓Cell proliferation, ↑Apoptosis	↓Proliferating cells ↑Apoptotic cells	Murine endometriosis model	[97]
**Rosmarinic acid**			
↓Cell proliferation, ↓Oxidation	Cell cycle arrest in the G2/M phase ↓ROS	Endometrial stromal T-HESC cell line Primary endometriotic stromal cells	[97]
↓Endometriotic tissue volume	↓PCNA positive cells, ↑Apoptotic cells in lesions	Murine endometriosis model	[97]
**Genistein and daidzein**			
↓Cell proliferation ↓Inflammation ↓Estrogen biosynthesis	NF-κB inhibition, ↓*IL6*, *↓IL8*, *↓COX2*, ↓PGE_2_ Higher affinity toward ERβ than ERα ↓*CYP19A1* ↓Aromatase activity ↓Glucocorticoid-regulated kinase in serum	Primary endometriotic stromal cells from ovarian endometrioma	[98]
↓Endometriotic lesions growth	↓Ki-67-positive cells	Murine endometriosis model	[98]
**Genistein**			
↓Endometriotic implant size	Antagonistic estrogen activity	Rat endometriosis model	[99]
**Puerarin**			
↓Cell invasion ↓Angiogenesis	↓MMP-9 ↑TIMP-1 ↓ICAM-1 ↓VEGF	Primary endometriotic stromal cells Chick chorioallantoic membrane model	[100]
↓Cell proliferation ↑Antiestrogen activity	↓*CCND1*, ↓Cyclin D1, ↓*CDC25A*, ↓Cdc25A ↑ERα corepressors (SMRT, NCoR) ↓ERα coactivators (SRC-1, SRC-3)	Endometriotic stromal cells	[101]
↑Antiestrogen activity	↓*CYP19A1*, ↓aromatase activity, *↓HSD17B1*, *↑HSD17B2,* ↓COX-2, *↓COX2,* ↓PGE_2_, ↓ERβ, ↓E2	Rat endometriotic model	[102]
**Naringenin**			
↓Cell proliferation ↑Apoptosis	PI3K and MAPK pathway activation ↓Mitochondrial membrane potential, ↓ROS	Endometriosis cell lines	[103]
↓Endometriotic lesions growth ↓Angiogenesis, ↓Inflammation ↑Apoptosis ↓Endometriosis prognostic markers	↓MMP-2, ↓MMP-9, ↓TNF-α, ↓NO, ↑ROS ↓Mitochondrial membrane potential, ↓VEGF ↓BCL-2, ↓PCNA, ↑Caspase-3, ↑Cyt-C ↓Nrf2/HO1/NQO1 signaling, ↓TAK1, ↓PAK1	Rat endometriosis model	[104]
**Xanthohumol**			
↓Endometriotic lesions growth ↓Angiogenesis	↓PI3K ↓Microvessel density	Murine endometriosis model	[105]
**Epigallocatechin gallate**			
↓Cell proliferation, ↓Cell migration ↓Cell invasion, ↓Fibrogenesis	↓MAPK signaling, ↓Smad signaling ↓*αSMA*, ↓*Col-I*, ↓*CTGF*, ↓*FN*	Endometriotic and endometrial stromal cells from patients	[106]
↓Endometriotic implant size ↓Angiogenesis	↓Microvessel number and size ↓*VEGFC/VEGFR2* expression and signalling	Murine endometriosis model	[107]
↓Endometriotic lesions growth ↑Apoptosis, ↓Angiogenesis	↓Lesion size and weight, ↓Serum VEGF ↑Apoptotic cells in lesions, ↓αSMA, ↓CD31	Murine endometriosis model	[108]
**Curcumin**			
↑Cell adhesion ↓Inflammation	↓*ICAM1*, ↓ICAM-1, ↓*VCAM1*, ↓VCAM-1 ↓IL-6, ↓IL-8, ↓MCP-1, NF-κB inhibition	Primary endometriotic stromal cells	[109]
↓Cell proliferation	↓E2 production	Primary endometriotic stromal cells	[110]
Cell cycle arrest ↑Apoptosis	↑Cells in the G1 phase, ↓Cells in the S phase ↓VEGF	Primary endometriotic and endometrial stromal cells	[111]
↓Inflammation ↓Angiogenesis	↓IL-6, ↓IL-8, ↓IP-10, ↓G-CSF, ↓MCP-1 ↓RANTES	Primary endometriotic stromal cells derived from eutopic endometrium	[112]
↑Apoptosis ↓Endometriotic lesions growth	Cytochrome c-mediated mitochondrial pathway modulation, p53-dependent and -independent pathway modulation, NF-κB inhibition, ↓MMP-3	Murine endometriosis model	[113]
↓Inflammation, ↓Oxidation	↓MMP-9, ↓TNF-α, ↓Lipid and protein oxidation	Murine endometriosis model	[114]
↓Endometriotic tissues weight and volume	↓VEGF	Rat endometriosis model	[115]
↓Angiogenesis ↓Endometriotic lesions growth	↓Microvessel density	Rat endometriosis model	[116]
**Resveratrol**			
↓Cell invasion	↓Cell invasion in Matrigel	Primary endometriotic stromal cells	[117]
Estrogenic activity	Dose-dependent agonistic and antagonistic activity	Endometrial Ishikawa cell line	[118]
↓Cell proliferation, ↑Apoptosis	↓Cell number, DNA fragmentation	Primary endometrial epithelial cells	[119]
↓Endometriotic lesion number and volume, ↓Cell proliferation ↑Apoptosis, ↓Angiogenesis	↓PCNA, ↓CD34, ↓Peritoneal VEGF ↓Vascular density area, DNA fragmentation	Murine endometriosis model	[119]
↓Cell proliferation, ↑Apoptosis	↓MKI67, ↑PCNA, DNA fragmentation	Murine endometriosis model	[117]
↓Cell proliferation	↓ERα, ↓Ki-67	Immunocompromised mouse endometriosis model (RAG-2 knockout)	[118]
↓Cell proliferation	↓IGF-1, ↓*HGF*, ↓HGF	Primary eutopic and ectopic endometrial stromal cells from endometriosis patients	[120]
↓Cell proliferation, ↑Apoptosis ↓Angiogenesis	↑*P53*, ↑*BAX*, ↑*CASP3*, ↑*SIRT1*, ↓NO	3D primary endometriotic and endometrial cell cultures	[121]
↓Inflammation	↓*MCP1*, ↓*IL6*, ↓*IL8*, ↓MCP-1, ↓IL-6, ↓IL-8, ↓RANTES	Primary ectopic endometrial stromal cells	[122]
↓Endometriotic lesion size ↓Angiogenesis, ↓Inflammation	↓VEGF in peritoneal fluid, plasma and tissue ↓MCP-1 in peritoneal fluid and plasma	Rat endometriosis model	[123]
↓Endometriotic lesion size ↓Angiogenesis, ↓Inflammation	↓VEGF in peritoneal fluid and endometriotic tissue. ↓MCP-1 in peritoneal fluid	Rat endometriosis model	[124]
↓Endometriotic lesion size ↓Oxidation	↑SOD activity in serum and tissue ↑GPX activity in serum and tissue	Rat endometriosis model	[125]
↓Cell proliferation, ↓Cell migration ↓Cell invasion	EMT process suppression ↓MTA1, ↓ZEB2	Primary endometrial stromal cells Murine endometriosis model	[126]
↓Endometriosis-related pain	↓aromatase activity, ↓COX-2	Patients treated for 2 months with 30 mg resveratrol to the hormone therapy	[127]
↓Cell invasion	↓*MMP2*, ↓*MMP9*, ↓MMP-2, ↓MMP-9 in endometrial tissue, fluid and serum	34 patients with endometriotic infertility treated with 400 mg resveratrol twice daily for 12–14 weeks with contraceptives	[128]
↓ Angiogenesis, ↓Inflammation	↓*VEGF*, ↓*TNF*, ↓VEGF, ↓TNF-α in the eutopic endometrial tissue	34 patients with endometriosis within the implantation window treated with 400 mg resveratrol for 12–14 weeks	[129]

↑ Increase; ↓ Decrease; the gene name is italicized, protein name is not italicized. Abbreviations used in Table 1: HSD17B1, HSD17B2, 17β-hydroxysteroid dehydrogenase type 1 and 2; AKT, protein kinase B (PKB); ATG5, autophagy protein 5; BAX, BCL-2 associated X protein; BCL-2, B-cell lymphoma 2; BECN1, Beclin 1; CA-125, mucin-16; CASP-3, caspase-3; CCND1, G1/S-specific cyclin-D1; CD31, platelet endothelial cell adhesion molecule; CDC25A, M-phase inducer phosphatase 1; Col- I, collagen type1; COX-2, cyclooxygenase-2; CTGF, connective tissue growth factor; CYP19A1, aromatase gene; Cyt-C, cytochrome c; E2, estradiol; ER stress, endoplasmic reticulum stress; ERK1/2, extracellular signal-regulated kinase 1/2; ERα, estrogen receptor alpha; ERβ, estrogen receptor beta; FN, fibronectin; FURIN, oaired basic amino acid cleaving enzyme (proprotein convertase of MMPs); G-CSF, granulocyte colony-stimulating factor; GPX, glutathione peroxidase; HGF, hepatocyte growth factor; HO1, heme oxygenase; ICAM-1, intercellular adhesion molecule 1; IGF-1, insulin-like growth factor-1; IL-6, interleukin 6; IL-8, interleukin 8; IP-10, interferon gamma-induced protein 10; JNK, c-Jun N-terminal kinases; MAPK, mitogen-activated protein kinase; MCP-1, monocyte chemoattractant protein 1; MKI67, marker of proliferation Ki-67; MMP-2, matrix metalloproteinase 2; MMP-9, matrix metalloproteinase 9; MTA1, metastasis-associated protein 1; mTOR, mammalian target of rapamycin; MT1-MMP, membrane type 1-matrix metalloproteinase; NCoR, nuclear receptor corepressor 1; NF-κB, nuclear factor kappa B; NO, nitric oxide; NQO1, NAD(P)H quinone oxidoreductase 1; NRF2, nuclear factor erythroid 2-related factor 2; P53, cellular tumor antigen p53; PAK1, p21-activated kinase 1; PCNA, proliferating cell nuclear antigen; PGE_2_, prostaglandin E_2_; PI3K, phosphatidylinositol-4,5-bisphosphate 3-kinase; RANTES, regulated on activation, normal T cell expressed and secreted; ROS, reactive oxygen species; SIRT1, sirtuin 1; SMRT, silencing mediator of retinoic acid and thyroid hormone receptor; SOD, superoxide dismutase; SRC-1, steroid receptor coactivator 1; SRC-3, steroid receptor coactivator 3; TAK1, transforming growth factor beta-activated kinase 1; TGFB1, transforming growth factor beta-1 proprotein; TIMP-1, tissue inhibitor of metalloproteinase 1; TNF-α, tumor necrosis factor alpha; VCAM-1, vascular cell adhesion protein 1; VEGF, vascular endothelial growth factor; VEGFC, vascular endothelial growth factor C; VEGFR2, vascular endothelial growth factor receptor 2; ZEB2, zinc finger E-box-binding homeobox 2; αSMA, α-Smooth muscle actin.

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
