# Peer review of "Polyphenols as a Diet Therapy Concept for Endometriosis—Current Opinion and Future Perspectives"

_nutrients, 2021, doi:10.3390/nu13041347_

Round 1

Reviewer 1 Report

  1. Minor English editing is advised since there are many non-American variants are used in the text. Take the abstract; for example, minimising should be minimizing. Besides,oestrogen should be estrogen.
  2. I suggest adding the following references and revise the manuscript accordingly since they are easily found but not cited in the current format.
  3. Antioxidant Supplementation Reduces Endometriosis Related Pelvic Pain in Humans. Nalini Santanam,1Nino Kavtaradze,2 Ana Murphy,3 Celia Dominguez,4 and Sampath Parthasarathy5,*Transl Res. 2013 Mar; 161(3): 189–195.
  4. Therapeutic Approaches of Resveratrol on Endometriosis via Anti-Inflammatory and Anti-Angiogenic Pathways. Ana-Maria Dull, Marius Alexandru Moga, Oana Gabriela Dimienescu,*Gabriela Sechel,* Victoria Burtea, and Costin Vlad Anastasiu  2019 Feb; 24(4): 667.Gynecol Obstet Invest (DOI:10.1159/000512628)
  5. Nonhormonal Treatment for Endometriosis Focusing on Redox Imbalance. Nagayasu M.a · Imanaka S.a,b · Kimura M.a · Maruyama S.a · Kobayashi H.a,b Gynecol Obstet Invest. 2021 Jan 4;1-12. doi: 10.1159/000512628. 

Reviewer 2 Report

The review comprehensively summarises the evidence on the possible implication of dietary polyphenols in endometriosis therapy. The first part of the review is well designed and organised, which ease the readability of the paper. However, in the second part there are some major issues regarding classification of polyphenols.

Major comments

Line 570/571 Curcumin is included in the following class of polyphenols “Other polyphenols”, while Rosmarinic acid is considered as “Phenolic acid”. Please see phenol-explorer database for the classification of polyphenols.

Line 343-348 please see phenol-explorer database for adequate classification polyphenols into 6 main classes. The text following the introduction to polyphenols should be organised accordingly, providing evidence for each class of polyphenols.

Minor comments

My overall comment based on this article is that polyphenols could play a significant role not only in the treatment of endometriosis but also in its onset, as dietary factors are among risk factors for this condition. This could be emphasised in the manuscript.

Moreover, it would be more suitable to refer to meta-analysis of observational studies when stating the health effects of polyphenols rather than mechanistic studies.

Line 329-331, page 7: Reference 30 and 78 are outdated and not entirely corresponding to the fact stated in this lines. Authors should rather discuss more general evidence from the observational studies which explore the impact of dietary polyphenols intake on cancer risk (PMID: 27943649) as well as survival and recurrence (PMID: 32632445). Considering that the effect of the individual polyphenols on the endometriosis related conditions was discussed in the second part of the review (I.e line 481, 424, 634), the results from this studies should be also discussed. Finally, as endometriosis is associated with higher risk of CVD (both conditions underlined by a low grade sub-clinical inflammation), the effect of dietary polyphenols toward CVD should be mentioned in an adequate paragraph (PMID: 33559970).

Round 2

Reviewer 2 Report

Authors addressed all the comments and improved polyphenols nomenclature/classification.